# Optimizing blood alcohol concentration measurement with breath alcohol meter and its preliminary application in evaluating CYP2E1 activity

Jiayi Zhang[☉], Yingqi Hu[☉], Ziqi Jin, Runa A, Xiaoxue Wang, Lingyu Zhang, Yongzhi Xue[iD]*

Department of Pharmacology, Baotou Medical College, Institute of Pharmacokinetics and Liver Molecular Pharmacology, Baotou, China

☉ These authors contributed equally to this work.
* xyzhxyzh68@sohu.com

## Abstract

### Background

Ethanol metabolism *in vivo* is primarily mediated by CYP2E1 and alcohol dehydrogenase, with CYP2E1 playing the predominant role at high ethanol concentrations. However, it is a difficult problem to evaluate CYP2E1 activity with ethanol as a probe, especially in rats. Currently, high-performance liquid chromatography (HPLC)-based detection of chlorzoxazone is the standard method for evaluating CYP2E1 activity. Nevertheless, this method is labor-intensive, time-consuming, costly, and unsuitable for real-time on-site or multipoint noninvasive monitoring.

### Objectives

This study aimed to develop and optimize a noninvasive and rapid method for blood alcohol concentration BAC detection in rats. The optimized approach was subsequently applied to preliminarily assess hepatic CYP2E1 activity.

### Methods

Male Sprague-Dawley rats were randomly divided into two groups: the immune-mediated liver injury (hepatitis) group and the control group. Hepatitis rats was induced by tail vein injection of Bacillus Calmette-Guerin and lipopolysaccharide, whereas the control rats received an equivalent volume of normal saline. On day 14 of the experiment, following intragastric administration of 56% (v/v) alcohol (5 mL·kg$^{-1}$), BAC in both groups was measured using a breath alcohol meter. The BAC-time curve was segmented at 46 mg·dL$^{-1}$: the upper portion was used to assess CYP2E1 activity, while the lower portion reflected alcohol dehydrogenase activity. Liver tissues were collected from rats and observed for histopathological changes

**Data availability statement:** All relevant data are within the paper and its Supporting Information files.

**Funding:** This work was supported partly by the national natural science foundation of China (No. 81460567, 82160709) and partly by Inner Mongolia natural science foundation of China (No.2014MS0813, 2019MS08198, 2023MS08066).

**Competing interests:** The authors declare that they have no competing interests.

using hematoxylin and eosin staining. To validate the accuracy of breath alcoholmeter, headspace gas chromatography was employed. Additionally, HPLC was used to determine plasma chlorzoxazone metabolism, serving as an independent measure of CYP2E1 activity. Based on these findings, we further investigated the potential of ethanol as a probe substrate for assessing the activity of CYP2E1.

## Results

The pathological findings demonstrated that BCG successfully induced immune liver injury in rats. Comparative analysis using both breath alcohol meter and gas chromatography revealed significantly reduced alcohol metabolism in the immune-mediated liver injury rats relative to the control. The gas chromatographic measurements of BAC confirmed the accuracy and reproducibility of the breath alcohol detection method. Furthermore, HPLC analysis demonstrated a marked reduction in CYP2E1 activity in the liver injury rats compared to control.

## Conclusion

The breath alcohol detection method offers a simple, non-invasive approach that can serve as a viable alternative for assessing hepatic CYP2E1 metabolic activity.

## 1. Background

CYP2E1, a key member of the cytochrome P450 enzyme family, plays a key role in the metabolism of both exogenous and endogenous compounds. Beyond genetic regulation, diseases like hepatitis can profoundly disrupt CYP2E1 expression and activity [1]. Alterations in CYP2E1 may lead to either an increase or decrease in the plasma concentrations of its substrate drugs, thereby influencing therapeutic efficacy or potentially inducing drug-drug interactions. Hepatitis-driven hepatocellular carcinoma accounts for approximately 70% of liver cancer cases. During the progression of hepatitis and hepatocellular carcinoma, CYP2E1 undergoes remodeling and actively participates in pathological processes [2,3]. The overexpression of CYP2E1 leads to reactive oxygen species (ROS) accumulation and/or activation of carcinogens, which represents a significant contributing factor in liver disease pathogenesis [4,5]. Consequently, monitoring CYP2E1 metabolic activity is essential for elucidating the mechanisms underlying liver disease development. Current gold-standard methodologies for quantifying CYP2E1 activity primarily rely on chromatographic techniques, particularly high-performance liquid chromatography (HPLC) and ultra-performance liquid chromatography-tandem mass spectrometry [6]. These techniques, however, are constrained by multiple limitations including (1) requirement for specific probe substrates, (2) complex operational protocols, (3) high-cost instrumentation, and (4) lengthy analytical processes. Breath alcohol meter have been routinely employed for decades in drunk-driving detection [7], providing a noninvasive approach for blood alcohol concentration (BAC) monitoring. However,

their application in measuring BAC in rodent models has remained technically challenging. Our prior work developed a breathalyzer-based method for rat blood alcohol measurement [8], but its syringe-dependent sampling introduced variability. We now report an optimized system incorporating automated aspiration, enabling more reliable alcohol detection and preliminary CYP2E1 activity assessment.

Ethanol metabolism in vivo is primarily mediated through two key enzymatic pathways: alcohol dehydrogenase (ADH) and CYP2E1 [9–11]. ADH, characterized by its low Km value (< 5 mM), rapidly reaches saturation when BAC exceed 15–20 mg% [12]. Under high alcohol exposure conditions, CYP2E1 - with its substantially higher Km (≈10 mM; 46 mg·dL⁻¹) – demonstrates markedly enhanced activity and becomes the predominant metabolic enzyme [13–15]. Accordingly, we established a novel probe-based approach by segmenting the BAC -time curve (obtained after alcohol gavage in rats) at the 46 mg·dL⁻¹ threshold: concentrations above this value reflect predominantly CYP2E1 activity, while lower concentrations represent ADH-mediated metabolism. This methodology was systematically validated through: (1) gas chromatography (GC) verification of breath alcohol measurement accuracy, and (2) HPLC-based quantification of chlorzoxazone metabolism to confirm ethanol's suitability as a CYP2E1 probe. This study pioneers the application of breath alcohol technology in scientific research, establishing a noninvasive, operationally simple, and field-deployable method for dynamic monitoring of CYP2E1 activity. Our approach provides a novel strategy for assessing the function of hepatic CYP2E1.

## 2. Methods

### 2.1. Materials

Male Sprague-Dawley (SD) rats, 8–9 weeks old, were obtained from Speifu Beijing Biotechnology Co., Ltd. (Animal License No. SCXK [Beijing] 2024−0001). All rats had free access to food and water, and were housed in a controlled environment with a temperature of approximately 25°C, humidity of approximately 50%, and a 12/12 h light/dark cycle. The experimental personnel were professionally trained and skilled in operations, making every effort to minimize the pain of rats. All animal experiments complied with the ARRIVE guidelines. The experimental protocol was approved by the Ethics Committee of Baotou Medical College (Approval No.: 2021−061). All procedures were performed in accordance with relevant guidelines and regulations. Rats were euthanized by sodium pentobarbital overdose (I.P. 200 mg·kg⁻¹) upon reaching predefined humane endpoints approved by the Ethics Committee. Including: *(1) >20% body weight loss within 48 hours; (2) severe lethargy or inability to reach food/water; (3) signs of irreversible distress (e.g., labored breathing, ulcerative tumors).* To capture rapid drug concentration changes, blood samples (0.1 mL each) were collected after injection via orbital puncture under isoflurane anesthesia (2–3%). And use lidocaine for eye pain relief. Total blood loss was limited to <1.2 mL (≤8% of total volume) over 9 hours. Between samples, rats recovered in warmed cages with free access to water. Among 32 rats (n = 8 per group), no premature mortalities occurred; all deaths followed the fulfillment of approved humane endpoints. Throughout the experimental period, all rats were subjected to twice – daily observations by trained researchers using a standardized scoring system (e.g., body condition score) at 08:00 and 16:00 hours.

Bacillus Calmette-Guerin (BCG) vaccine (batch number: 20231215) were obtained from Shanghai Ruichu Biotechnology Co., Ltd., lipopolysaccharide (batch number: 20240328) were obtained from Beijing Solarbio Technology Co., Ltd., chlorzoxazone (batch number: 481850) were obtained from Beijing Bellwether Technology Co., Ltd., 6-hydroxychlorzoxazone (batch number: YJ10587) were obtained from Shanghai Yiji Industry Co., Ltd., β – glucosidase (batch number: JS248107) were obtained from Shanghai Yuanye Biotechnology Co., Ltd., 56% (v/v) Erguotou Baijiu (, batch number: 202205054) were obtained from Beijing Shunxin Agriculture Co., Ltd.

Black Cat No. 3 Breath Alcohol meter were obtained from Shenzhen Zhaowei Technology Co., Ltd., Gas Collecting Cylinder (3L), GC126 Gas Chromatography (DK-3001N Headspace Sampler, SPH-300 Hydrogen Generator, and SPB-3 Fully Automatic Air Source) were obtained from Beijing Zhonghui Analytical Technology Research Institute;

 

high-performance liquid chromatography system (1525 pump, 2998 diode array UV visible detector, USA) were obtained from Waters Technology Shanghai Co., Ltd.

## 2.2. Grouping of rats

Throughout the study, rats were maintained under standard laboratory conditions (22 ± 2°C, 55 ± 5% humidity) with ad libitum access to food and water. Following a 7-day acclimatization period, animals were randomly divided into two groups: (1) the BCG group (n = 16) receiving 100 mg·kg⁻¹ BCG via tail vein injection to induce immune-mediated liver injury, and (2) the control group (n = 16) administered equivalent volumes of sterile saline. On day 13 post-injection, BCG-treated rats received 40 µg·kg⁻¹ lipopolysaccharide via tail vein while controls received saline. For metabolic assessments on day 14, half of each group (n = 8) underwent alcohol pharmacokinetic studies involving oral administration of 56% (v/v) alcohol (5 mL·kg⁻¹) with serial blood collection via the orbital plexus at 20, 40, 60, 120, 180, 300, 420, 540 and 660 min post-dosing for GC analysis (Section 2.4.3), while the remaining animals received 80 mg·kg⁻¹ chlorzoxazone by gavage with blood sampling at 5, 10, 30, 60, 120, 180, 240, 360 and 600 min for HPLC-based CYP2E1 activity assessment (Section 2.5.3).

Liver tissue samples were collected from both control and BCG rats. The tissues were fixed in a pre-prepared formaldehyde-alcohol fixative (formaldehyde: absolute ethanol = 1:9) for 48 hours, followed by paraffin embedding. Sections were stained with hematoxylin and eosin (H&E) for histological examination. Digital whole-slide images were acquired using a Hamamatsu pathology biopsy scanner (Hamamatsu Photonics, Tokyo, Japan). Histopathological evaluation was performed under the supervision of experienced pathologists.

## 2.3 Measurement of BAC Using a Breath Alcohol Meter

On the 14th day, the BAC of rats was monitored after intragastric administration of 56% alcohol (5 mL·kg⁻¹) using a breath alcohol meter. Measurements were taken at 20-minute intervals from 20 to 540 minutes post-administration. The detection method was optimized based on our laboratory's previous protocol [8]. Before each measurement, the rat was placed in a gas collection chamber for 10 min. After this period, the breath alcohol meter was positioned at the chamber's opening to automatically detect the breath alcohol concentration and convert it into BAC. This procedure was continued for up to 9 h until the breath alcohol concentration reached zero. A time-dependent BAC curve was plotted.

## 2.4 Determination of BAC by GC

**2.4.1 Chromatographic conditions.** The analysis was performed following a previously described method [14] with modifications. A DB-ALC capillary column (130 m × 0.32 mm × 1.8 µm) was used for separation. The temperature program was set as follows: initial temperature at 120 °C (held for 5 min), followed by a ramp at 6 °C·min⁻¹ to 200 °C (held for 2 min). The injector and detector temperatures were maintained at 230 °C and 300 °C, respectively. Nitrogen was employed as the carrier gas at a flow rate of 3 mL·min⁻¹ with a split ratio of 5:1.

**2.4.2 Standard curve.** A series of ethanol standard solutions were prepared at concentrations of 0.625, 1.25, 2.5, 5, 10, 20, 40, 80, 160, and 320 mg·100mL⁻¹. Linear regression analysis was performed using the peak area (Y) versus concentration (X), yielding the calibration equation: $Y = 52.937X + 15.948$ ($R^2 = 0.9997$). The method exhibited excellent linearity within the range of 1.25–320 mg·100mL⁻¹, with a lower detection limit of 1.25 mg·100mL⁻¹, confirming its suitability for subsequent experiments.

**2.4.3 Plasma sample preparation.** For analysis, 100 µL of plasma was transferred into a crimp-sealed vial and placed onto the autosampler tray of the headspace GC for injection.

## 2.5 Determination of Chlorzoxazone Concentration by HPLC

**2.5.1 Chromatographic conditions.** The analytical method was established based on previous reports [15,16]. A C18 HPLC column (4.6 mm × 250 mm, 5 µm) was used with a mobile phase consisting of methanol: water (60:40, v/v) at a flow

rate of 0.8 mL·min$^{-1}$. The detection wavelengths for chlorzoxazone and 6-hydroxychlorzoxazone were set at 280 nm and 296 nm, respectively. The column temperature was maintained at 30 °C.

**2.5.2 Standard curves.** The solutions of chlorzoxazone were diluted to concentrations of 0.5, 2.5, 12.5, 25, 50, 100, 200, and 400 µg·mL$^{-1}$, while 6-hydroxychlorzoxazone were diluted to 5, 10, 20, 40, 80, and 160 µg·mL$^{-1}$. The samples were pretreated according to the procedure described in Section 2.5.3 and then analyzed by HPLC. The peak area (Y) was plotted against the concentration (X) to establish the linear regression equations. For chlorzoxazone, the regression equation was Y = 47,974X – 43,655 (R² = 0.9999), demonstrating excellent linearity over the range of 0.5–400 µg·mL$^{-1}$. For 6-hydroxychlorzoxazone, the regression equation was Y = 9,197.4X – 5,380.7 (R² = 0.9998), with a linear range of 2.5–40 µg·mL$^{-1}$.

**2.5.3 Plasma sample pretreatment.** To 100 µL of plasma, 200 µL of β-glucuronidase-containing saline and 0.8 mL of sodium acetate buffer were added, followed by incubation at 37 °C for 3 h. Subsequently, 1 mL of saturated sodium sulfate and 2 mL of ethyl acetate were added for liquid-liquid extraction. After vortex mixing for 10 min, the sample was centrifuged at 12,000 rpm (4 °C) for 10 min. The supernatant was collected and evaporated to dryness under nitrogen. The residue was reconstituted in the mobile phase, vortexed for 10 min, and centrifuged again under identical conditions (12,000 rpm, 4 °C, 10 min). The resulting supernatant was subjected to HPLC analysis.

## 2.6. Statistical analyses

All data are presented as mean ± SD. Pharmacokinetic parameters were derived through non-compartmental analysis in DAS 3.0. Differences between groups were assessed by Student's t-test in GraphPad Prism 8.0.2, with statistical significance defined as *p < 0.05 and **p < 0.01. Normality was assessed using the Shapiro-Wilk test, with a significance level of P > 0.05 indicating a normal distribution. Homogeneity of variances was evaluated with Levene's test, where P > 0.05 was considered to indicate equal variances between groups.

## 3. Results

### 3.1 Optimization of operation method for breath alcohol meter, and investigation on reproducibility and linearity

A breath alcohol meter with an automatic suction pump (Hunter-800, Shenzhen, China) was used to optimize the previous method of extracting alcohol gas with a syringe, which reduced operational errors (Fig 1A). Specifically, 1 mL of alcohol with a concentration of 1.6% (v/v) was added to a bottle cap with an inner diameter of 2.5 cm and placed at the central bottom of a gas cylinder. Data were statistically analyzed through five repetitions of the same procedure, and the coefficient of variation (CV) was calculated as 4.7% by dividing the standard deviation by the mean to investigate the method's reproducibility (Fig 1B). The results showed that the method was reproducible with minimal errors. Furthermore, alcohols

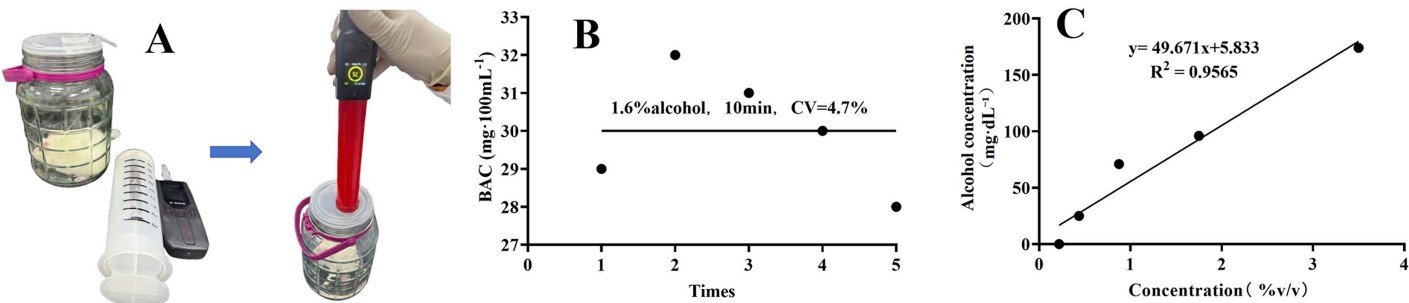

**Fig 1. Optimization of operation method for breath alcohol meter, and investigations on reproducibility and linearity.** (A) Optimization of the method; (B) Coefficient of variation; (C) Concentration dependency.

with different concentrations were placed in the gas cylinder for 10 min for detection. A scatter plot was drawn, followed by linear regression analysis, yielding a regression equation with an $R^2$ value of 0.9565 (Fig 1C), indicating an excellent linear relationship.

### 3.2 Histopathological Analysis of the Rat Liver

Histopathological examination revealed no abnormalities in control rats, whereas BCG rats exhibited histological features indicative of hepatitis. Liver sections stained with H&E were evaluated under light microscopy (Fig 2). In the control group, the hepatic lobule architecture remained intact, with radially arranged hepatic cords, uniformly sized hepatocytes, round nuclei, and minimal inflammatory cell infiltration in the portal areas (Fig 2A). In contrast, the BCG group showed extensive infiltration of mononuclear cells and lymphocytes within both the hepatic parenchyma and portal areas, forming diffuse inflammatory foci of varying sizes. These inflammatory aggregates resulted in disruption of the lobular architecture, as indicated by arrows in Fig 2B. A magnified view of the area outlined in Fig 2B is presented in Fig 2C, confirming the success of a BCG-induced hepatitis in rats.

### 3.3 Detection of Alcohol Metabolic Activity in BCG-Induced Hepatitis Using Breath Alcohol Meter

The effects of different liver pathophysiological states on alcohol metabolic activity were investigated by measuring the alcohol metabolism in control rats and BCG-induced hepatitis rats. Alcohol metabolism was significantly slower in the BCG group, as evidenced by the marked increase in BAC at 40, 220, 240, 260, and 280 min post-alcohol administration on the BAC-time curve (Fig 3A), along with a significant increase in $AUC_{(0-t)}$ (Fig 3B). The $AUC^{up}_{(0-t)}$ (area under the curve where BAC exceeded 46 mg·dL$^{-1}$) represents the metabolic fraction via CYP2E1. Notably, BCG-treated rats showed a significant increase in $AUC^{up}_{(0-t)}$, indicating slowed CYP2E1 metabolic activity (Fig 3C).

### 3.4 Detection of Alcohol Metabolic Activity in BCG-Induced Hepatitis by GC

To validate the feasibility and scientificity of the breath alcohol detection method, GC was used to detect alcohol metabolic activity in BCG-induced hepatitis. The results showed that alcohol metabolism was significantly slower in the BCG group, as demonstrated by the remarkable increase in BAC at 120, 180, 300, and 420 min post-alcohol administration on the BAC-time curve (Fig 4A), along with a significant increase in $AUC_{(0-t)}$ (Fig 4B). The $AUC^{up}_{(0-t)}$ (area under the curve where BAC exceeded 46 mg·dL$^{-1}$) representing the CYP2E1-mediated metabolic fraction was significantly higher in BCG-treated rats, indicating decreased CYP2E1 metabolic activity (Fig 4C).

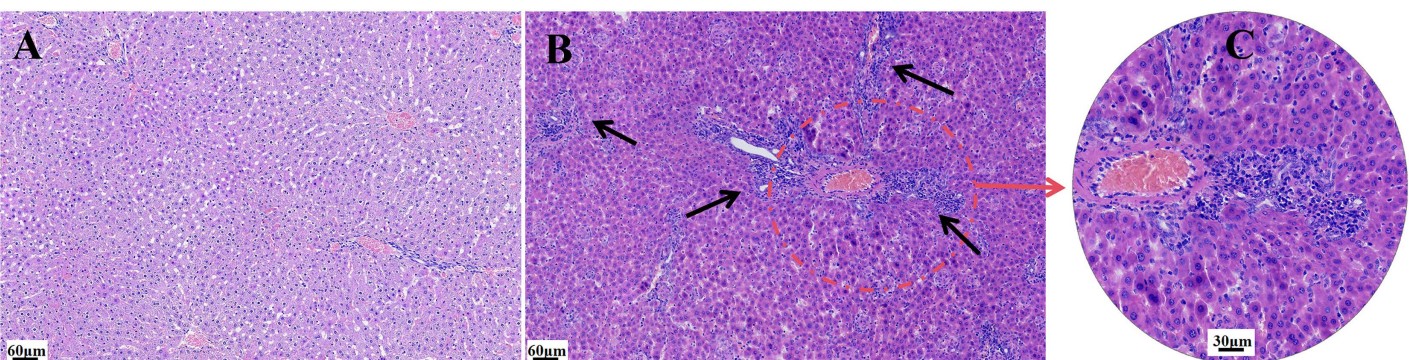

**Fig 2. Histopathological changes in liver tissues of rats.** (A) Control group (H&E staining; magnification ×200); (B) BCG-induced hepatitis group (arrows indicate inflammatory foci); (C) Higher-magnification view of the area marked in B.

### 3.5 Detection of CYP2E1 Metabolic Activity in BCG-Induced Hepatitis by HPLC

To validate the feasibility and scientificity of using alcohol as a probe to assess CYP2E1 metabolic activity via breath alcohol meter, HPLC was employed to detect the probe drug chlorzoxazone and its hydroxy metabolite in BCG-induced hepatitis. The results showed that chlorzoxazone metabolism was significantly slower in the BCG group, as indicated by the marked increase in chlorzoxazone plasma concentration at 60, 120, 180, 240, and 360 min post-administration on the BAC-time curve (Fig 5A), along with a significant increase in AUC (0-t) (Fig 5B). In contrast, the BCG group exhibited reduced production of the metabolite 6-hydroxychlorzoxazone, with significantly decreased plasma concentrations at 20, 40, and 60 min post-administration on the BAC-time curve (Fig 5C) and a significant reduction in AUC (0-t) (Fig 5D).

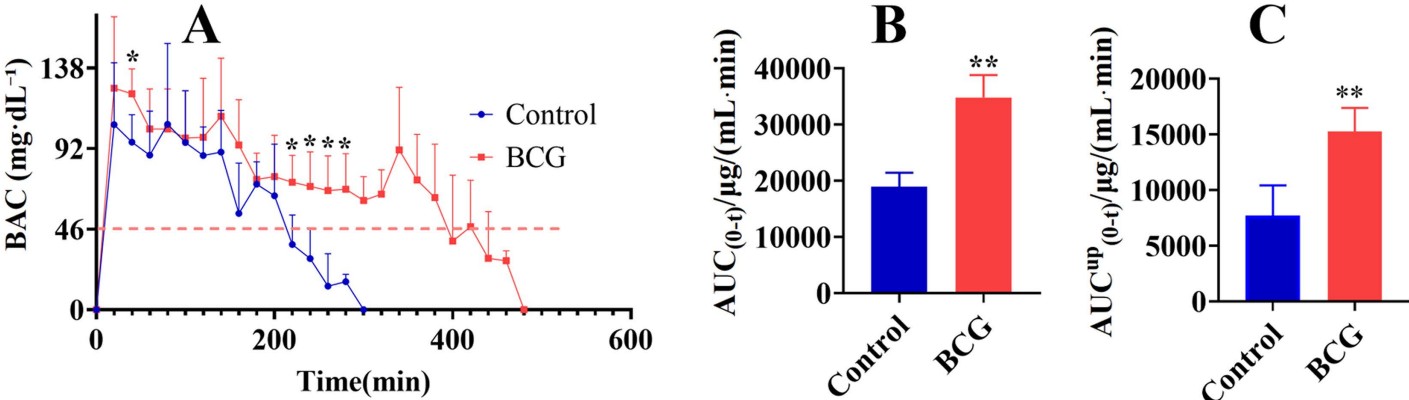

**Fig 3. Detection of alcohol metabolic activity in Bacillus Calmette-Guerin(BCG)-induced hepatitis using breath alcohol meter.** (A) Time-course changes of BAC in control and BCG rats; (B) AUC $_{(0-t)}$ in control and BCG rats. (C) AUC$^{up}_{(0-t)}$ in control and BCG rats. All data are presented as mean±standard deviation (n=5). *P<0.05, **P<0.01 vs. control group.

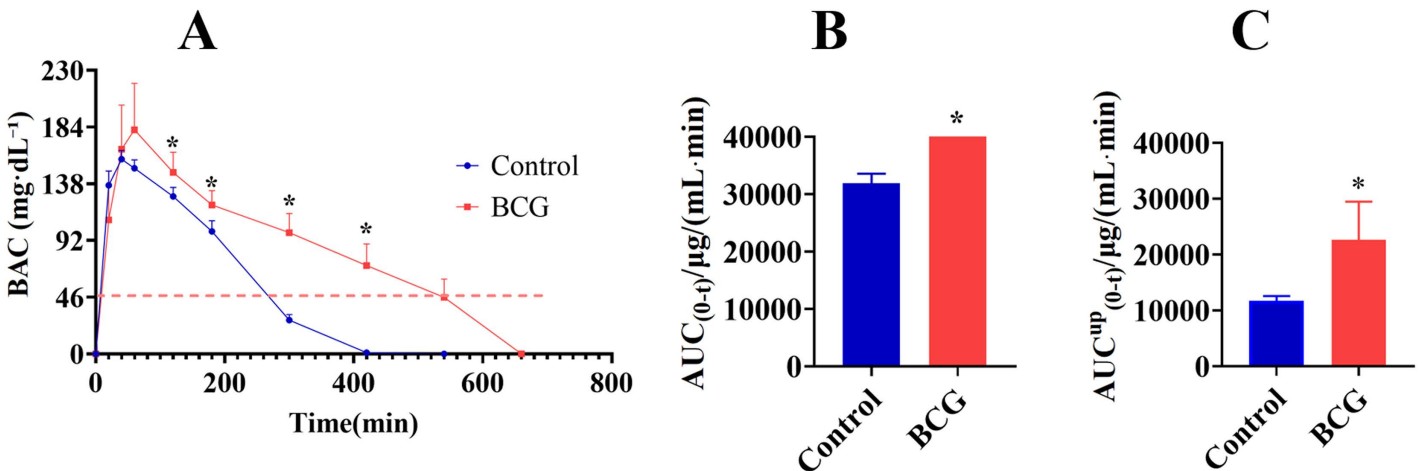

**Fig 4. Detection of alcohol metabolic activity in BCG-induced hepatitis by gas chromatography.** (A) Time-course changes of BAC in control and BCG rats. (B) AUC (0-t) in control and BCG rats. (C) AUC$^{up}_{(0-t)}$ in control and BCG rats. All data are presented as mean±standard deviation (n=5). *P<0.05, **P<0.01 vs. control group.

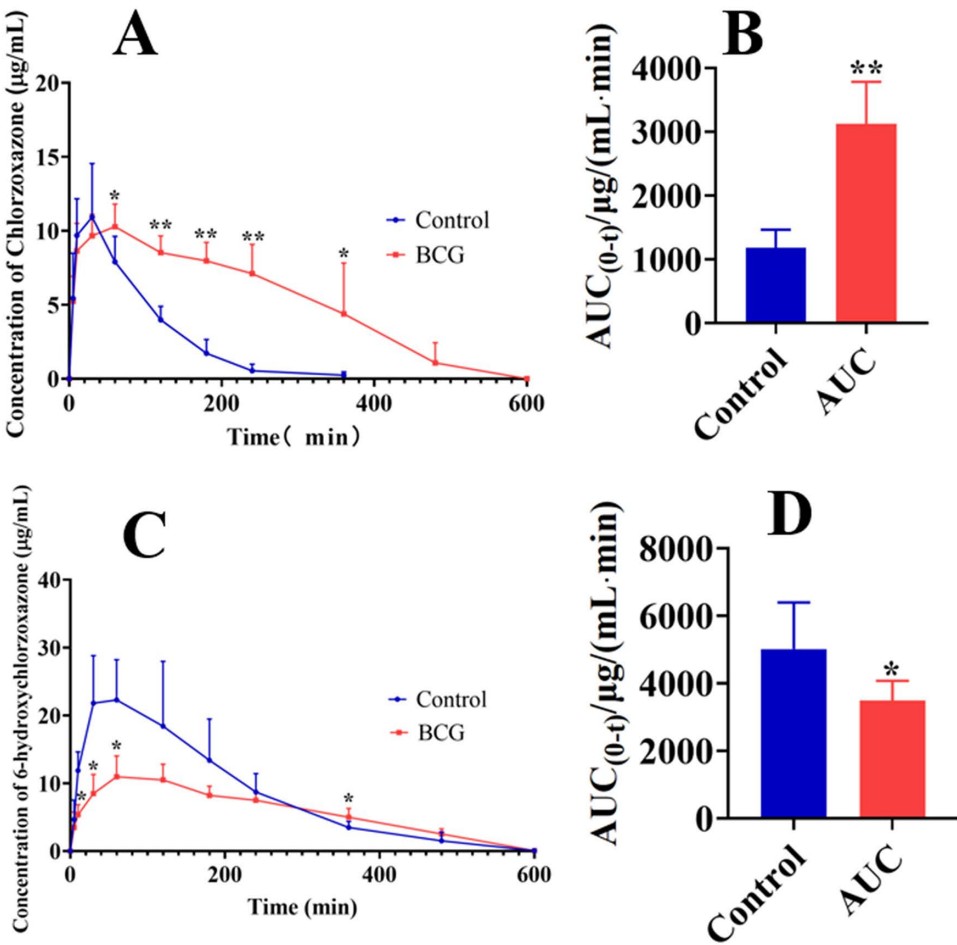

**Fig 5. Detection of CYP2E1 metabolic activity in BCG-induced hepatitis by high-performance liquid chromatography (HPLC).** (A) Time-course changes of chlorzoxazone plasma concentration in control and BCG rats. (B) Chlorzoxazone AUC $_{(0-t)}$ in control and BCG rats. (C) Time-course changes of 6-hydroxychlorzoxazone plasma concentration in control and BCG rats. (D) 6-Hydroxychlorzoxazone AUC $_{(0-t)}$ in control and BCG rats. All data are presented as mean±standard deviation (n=5). *P<0.05, **P<0.01 vs. control group.

## 4. Discussion

CYP2E1, a key member of the cytochrome P450 enzyme family, is involved in the metabolism of endogenous substances (such as acetone and fatty acids) and exogenous compounds, including anesthetics, ethanol, nicotine, acetaminophen, and the antiepileptic drug phenobarbital. Additionally, CYP2E1 activates toxic and carcinogenic compounds found in tobacco smoke and nitrosamines, contributing to disease development. Thus, the detection of in vivo CYP2E1 metabolic activity is frequently required [15–18].

In the present study, we optimized our previously established methods [8,15] by incorporating a breath alcohol meter equipped with an aspiration pump and refining the computing method. In earlier approaches, the BAC–time curve used a cutoff of 20 mg·dL $^{-1}$ to distinguish between ADH and CYP2E1 metabolic activities. However, ADH involvement in alcohol metabolism persists within the range of 20–46 mg·dL $^{-1}$, complicating the interpretation of CYP2E1 activity. According to previous reports, the Km value of CYP2E1 is approximately 10 mM [13–15], which corresponds to 46 mg·dL $^{-1}$. Therefore, using a cutoff of 46 mg·dL $^{-1}$ provides a more scientifically justified threshold for specifically assessing CYP2E1-mediated metabolism, as alcohol concentrations above this level predominantly reflect CYP2E1 activity.

To validate our approach, we replicated the BCG-induced hepatitis rat model, which has been successfully employed in several previous studies from our research team [5,15,18,19]. Prior work has confirmed that rats in the hepatitis group exhibit reduced hepatic CYP2E1 protein expression, metabolic activity, and transcriptional levels compared with control rats. In the current study, histopathological examination confirmed the successful establishment of the hepatitis model, supporting its suitability for subsequent experimental investigations.

Using this optimized method, we found that CYP2E1 metabolic activity was reduced in BCG-induced hepatitis. The portion of the BAC-time curve above 46 mg·dL$^{-1}$, primarily metabolized by CYP2E1, was used to assess CYP2E1 activity. Results showed that CYP2E1 metabolic activity in BCG-induced hepatitis was reduced to half that of the control group (calculated by area under the curve), indicating impaired metabolic function. Validation by headspace GC confirmed that alcohol metabolism and CYP2E1 activity were both reduced to half of the control levels in BCG-treated rats, consistent with the breath alcohol meter results. Furthermore, HPLC analysis of 6-hydroxychlorzoxazone metabolism showed that CYP2E1 activity in BCG-treated rats decreased to 3/5 of the normal level, aligning with the breath alcohol meter findings.

This original experimental method offers distinct advantages: Non-invasive dynamic monitoring via breath sample collection; Operational simplicity with minimal time consumption and potential for scientific translation; High consistency between detection results and classic enzyme activity assays.

The method enables non-invasive multi-point detection of CYP2E1 metabolic activity in rats, making it widely applicable to experimental studies of alcoholic and non-alcoholic liver injury. Alcohol is a popular beverage often consumed during meals due to its stimulatory effects on the central nervous system and ability to enhance social interactions [8]. Our computational approach, combined with breath alcohol monitoring, offers a non-invasive method to assess an individual's CYP2E1 metabolic activity. Furthermore, by analyzing the BAC-time curve, this strategy allows for estimation of the time required for complete alcohol clearance. Such an assessment is practically valuable for predicting when individuals can safely resume tasks requiring high precision, such as driving, after alcohol consumption. Additionally, this approach may serve as a biomarker for evaluating liver function [8,15,19]. Notably, a sufficient alcohol dosage is required, as CYP2E1-mediated metabolism predominates when BAC exceeds 46 mg·dL$^{-1}$. Moreover, time-course measurements of BAC are necessary to identify impaired alcohol metabolism by breath alcohol meter. Essentially, breath alcohol meter as tools for detecting BAC rather than directly measuring enzyme activity. This inherent limitation, associated with indirect detection, is also present in the conventional chlorzoxazone-based method. Such constraints can be mitigated through procedural optimizations to reduce indirect errors, supplemented by techniques such as measuring CYP2E1 protein and mRNA expression. Ultimately, this approach facilitates the establishment of a "rapid screening (breath alcohol meter) + precise verification (direct techniques)" framework. Such a system not only enables on-site identification of individuals with impaired alcohol metabolism but also clarifies the key role of CYP2E1 activity in the process. Despite the limitations noted above, the methodology established in this study offers a novel and non-invasive approach for the dynamic monitoring of CYP2E1 metabolic activity, thereby opening new avenues for further research and application.

## Supporting information

**S1 Data. Data.**
(XLSX)

## Acknowledgments

We would like to thank Professor Z GL from the Peking University School of Medicine for his technical support.

## Author contributions

**Conceptualization:** Jiayi Zhang, Yingqi Hu, yongzhi xue.

**Data curation:** Jiayi Zhang, Yingqi Hu, Ziqi Jin, runa a, Xiaoxue Wang, Lingyu Zhang, yongzhi xue.

**Formal analysis:** Jiayi Zhang, Yingqi Hu, runa a, Xiaoxue Wang, Lingyu Zhang, yongzhi xue.

**Funding acquisition:** yongzhi xue.

**Investigation:** Yingqi Hu, Ziqi Jin, yongzhi xue.

**Methodology:** Jiayi Zhang, Yingqi Hu, Ziqi Jin, runa a, Xiaoxue Wang, Lingyu Zhang, yongzhi xue.

**Project administration:** yongzhi xue.

**Resources:** yongzhi xue.

**Software:** Jiayi Zhang, Yingqi Hu, yongzhi xue.

**Supervision:** Ziqi Jin, yongzhi xue.

**Validation:** yongzhi xue.

**Visualization:** Ziqi Jin, yongzhi xue.

**Writing – original draft:** Jiayi Zhang, Yingqi Hu, Ziqi Jin, yongzhi xue.

**Writing – review & editing:** yongzhi xue.

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
