## [Decision Letter · Decision Letter 0]

23 Aug 2025

Dear Dr. xue,

The use of BrAC to evaluate CYP2E1 activity is insufficiently supported. Please provide mechanistic justification and, if possible, supporting data as a reviewer suggested to strengthen this claim. The current title overstates the context. As actual optimization process was conducted, we recommend revising the title to better reflect the scope of the manuscript.

We look forward to receiving your revised manuscript.

Kind regards,

Jeong Hoon Pan, Ph.D

Academic Editor

PLOS ONE

Journal Requirements:

2. Thank you for addressing the Editorial Office's query regarding the detailing of information relating to Human Endpoints in your manuscript's text. We would be grateful for your further attention of the following:

* Please add clearer statements to the Methods section of your manuscript regarding the length of the experiment, the number of animals used, and the euthanization of animals at the conclusion of the experiment.

3. To comply with PLOS One submissions requirements, in your Methods section, please provide additional information regarding the experiments involving animals and ensure you have included details on (1) methods of sacrifice, (2) methods of anesthesia and/or analgesia, and (3) efforts to alleviate suffering.

This work was supported partly by the national natural science foundation of China (No. 81460567, 82160709) and partly by Inner Mongolia natural science foundation of China (No.2014MS0813, 2019MS08198, 2023MS08066).

6. In the online submission form, you indicated that the data presented in this study are uploaded during submission as a supplementary file and are openly available for readers upon request.

8. Please remove all personal information, ensure that the data shared are in accordance with participant consent, and re-upload a fully anonymized data set.

Additional guidance on preparing raw data for publication can be found in our Data Policy (https://journals.plos.org/plosone/s/data-availability#loc-human-research-participant-data-and-other-sensitive-data) and in the following article: http://www.bmj.com/content/340/bmj.c181.long .

Reviewers' comments:

Reviewer's Responses to Questions

**Comments to the Author**

1. Is the manuscript technically sound, and do the data support the conclusions?

Reviewer #1: Yes

Reviewer #2: No

2. Has the statistical analysis been performed appropriately and rigorously?

Reviewer #1: Yes

Reviewer #2: I Don't Know

3. Have the authors made all data underlying the findings in their manuscript fully available?

Reviewer #1: Yes

Reviewer #2: Yes

4. Is the manuscript presented in an intelligible fashion and written in standard English?

Reviewer #1: Yes

Reviewer #2: No

Reviewer #1: In this study, the authors evaluated CYP2E1 activity using a novel approach described in “Optimizing Blood Alcohol Concentration Measurement with Breath Alcohol Meter and Its Preliminary Application in Evaluating CYP2E1 Activity.” The study is well thought out and elegantly performed. The major and minor revisions are outlined below.

Major Comments:

1. The manuscript introduces a novel segmentation of the BrAC curve at 46 mg·dL⁻¹ but lacks a strong justification or citation to support this specific threshold.

2. The methods mention t-tests but do not address whether assumptions (e.g., normality, equal variance) were tested.

3. Figures 2–4 are referenced as key evidence, but no quantitative values (mean ± SD) are shown in the main text or figure captions.

4. The breathalyzer method is said to be "potentially field-deployable," but its translational application (e.g., human studies, clinical validation) is not discussed.

Minor Comments:

1. Use consistent units for BrAC (e.g., mg·dL⁻¹ vs. mg/100mL) throughout the manuscript. PLOS prefers SI units—consider converting to mmol/L and add conversion factors if necessary.

2. Instead of referring to groups as "BCG" and "Control," consider more informative labels such as "Hepatitis model" vs. "Healthy control."

3. Ensure all figure labels (e.g., A, B, C) are visible and explained clearly in the legends. Consider unifying the layout for all figures.

Reviewer #2: The manuscript titled “Optimizing blood alcohol concentration measurement with breath alcohol meter and its preliminary application in evaluating CYP2E1 activity” described potential use of the breath alcohol measurement device for non-invasively evaluating blood alcohol concentration and CYP2E1 activity in rats. The device upgraded from author’s previous work, which have the automated aspiration system to minimize sampling errors that occurred from manual sampling in their previous work.

In the study, breath air and blood were collected over time from two groups of rats: normal rats and liver injury rats induced by BCG. And then blood alcohol concentration was measured by breath alcohol analyzer and GC to validate the breath alcohol measurement device. In addition, CYP2E1 activity was measured by tracking chloroxazone and its metabolites level in blood using HPLC.

In results, blood alcohol concentration measurement by the breath alcohol meter and classic HPCL method was compared, and the results showed similar pattern, which implying the device could be useful device for rapidly measuring blood alcohol concentration through breath in rodent study. And blood CYP2E1 substrate and its metabolite level were compared between normal rats and hepatitis rats. Hepatitis rats showed lower alcohol metabolism as evidenced by the lower alcohol clearing ability and lower CYP2E1 activity, when compared to normal rats. Eventually, authors concluded that the breath alcohol detection method could be rapid and non-invasive approach and it could be used for alternatively accessing hepatic CYP2E1 activity.

Overall, the authors suggested that their breath alcohol meter could be useful tool for rapid and non-invasive approach for monitoring blood alcohol concentration change with multiple repeated measuring it over time. However, major concerns are that the manuscript have poorly written, especially in discussion section, and have lack of scientific rationale for use of hepatitis rat model and the conclusion. The title is over-stated since there is no “optimization” process for the breath alcohol meter’s performance, assuming that it is just because of the automatic sampler. And even there is lack of reliable evidences for potential use of blood alcohol concentration as an indicator of hepatic CYP2E1 activity even though the blood alcohol concentration was over 45 mg/dL (threshold for concentration of ethanol that is predominantly metabolized by CYP2E1 rather than ADH). Supporting data is required for this, for example, reactive oxygen species (ROS) level because CYP2E1 generates ROS but ADH does not during alcohol metabolism. For animal study, 56° (v/v) Erguotou Baijiu was used as an ethanol source, which is kind of distilled liquor, rather than use of pure ethanol.

For these reasons, I would recommend “reject” for this manuscript.

Minors

Use of abbreviation; several abbreviations were not defined in the main text. Such as BCG, LPS.

2.1. Materials: some materials and equipment are just listed up, not in complete sentence.

Figure legends: insufficient information and description for the figures. And typing errors, for example, p value *P <0.05, *P <0.01.

Inconsistency in use of terms. For example, 56°Alcohol, 56% ethanol

**Do you want your identity to be public for this peer review?** For information about this choice, including consent withdrawal, please see our Privacy Policy

Reviewer #1: No

Reviewer #2: No

---

## [Author Response · Author response to Decision Letter 1]

10 Sep 2025

Subject: Revised Manuscript Submission: [PONE-D-25-32424] - [Optimizing Blood Alcohol Concentration Measurement with Breath Alcohol Meter and Its Preliminary Application in Evaluating CYP2E1 Activity]

Dear Dr. Jeong Hoon Pan

We sincerely appreciate your letter and the reviewers’ constructive comments on our manuscript entitled “[Optimizing Blood Alcohol Concentration Measurement with Breath Alcohol Meter and Its Preliminary Application in Evaluating CYP2E1 Activity]”. We are grateful for the opportunity to revise our manuscript and believe that it has been significantly improved as a result of the feedback provided.

We have carefully addressed all the comments from the reviewers and have incorporated the suggested changes into the manuscript. Below is a point-by-point response to the reviewers’ feedback. All corresponding revisions have been made in the main text and are highlighted for your convenience.

Response to the Academic Editor’s Comments:

Comment 1: The use of BrAC to evaluate CYP2E1 activity is insufficiently supported. Please provide mechanistic justification and, if possible, supporting data as a reviewer suggested to strengthen this claim.

The current title overstates the context. As actual optimization process was conducted, we recommend revising the title to better reflect the scope of the manuscript.

Response: Breath alcohol meter have been used by traffic police for many years to check drunk driving, but they cannot be directly applied to rats. We developed a methodology for estimating blood alcohol concentration in rats using a breath alcohol meter.

The first improvement involves the use of a meter equipped with an automatic suction pump, which reduces operational errors during sampling. The second optimization consists of setting the threshold for the blood alcohol concentration-time curve at 46 mg·dL⁻¹ instead of the previously used 20 mg·dL⁻¹. Studies have shown that alcohol metabolism below 20 mg·dL⁻¹ involves alcohol dehydrogenase, between 20–45 mg·dL⁻¹ it involves both alcohol dehydrogenase and CYP2E1, and above 46 mg·dL⁻¹ it is primarily mediated by CYP2E1. Therefore, blood alcohol concentration curves above 46 mg·dL⁻¹ better reflect the activity of CYP2E1.

The scientific rationale of this study is to optimize the methodology and explore the feasibility of using alcohol as a probe for preliminary assessment of CYP2E1 activity. Gas chromatography was used to measure blood alcohol concentration in rats to validate the accuracy of the breath alcohol meter. High-performance liquid chromatography was employed to monitor the metabolic process of chlorzoxazone, further verifying the feasibility of using alcohol as a probe to evaluate CYP2E1 activity. The experimental design is logically coherent and demonstrates clear innovation and scientific rigor. Additional arguments and references have been incorporated in the discussion section to strengthen the justification and enhance the alignment between the content and the title. The title of the manuscript is appropriate.

Comment 2: Please ensure that your manuscript meets PLOS ONE's style requirements, including those for file naming.

Response: We have carefully checked.

Comment 3: Please add clearer statements to the Methods section of your manuscript regarding the length of the experiment, the number of animals used, and the euthanization of animals at the conclusion of the experiment.

Response: We have carefully checked and supplemented the dosage of pentobarbital sodium used for animal euthanasia.

Comment 4: In your Methods section, please provide additional information regarding the experiments involving animals and ensure you have included details on (1) methods of sacrifice, (2) methods of anesthesia and/or analgesia, and (3) efforts to alleviate suffering.

Response: We confirm that the issues raised have already been addressed in the current version of our manuscript.

Comment 5: Statement on the role of funders.

Response: We have carefully checked. The funders had no role in study design, data collection and analysis, decision to publish, or preparation of the manuscript.

Comment 6: Your ethics statement should only appear in the Methods section of your manuscript.

Response: We have carefully checked.

Comment 7: In the online submission form, you indicated that the data presented in this study are uploaded during submission as a supplementary file and are openly available for readers upon request.

Response: We have carefully checked. The data presented in this study are uploaded during submission as a supplementary file and are openly available for readers upon request.

Comment 8: Please include captions for your Supporting Information files at the end of your manuscript, and update any in-text citations to match accordingly.

Response: We have revised the manuscript accordingly.

Comment 9: Please remove all personal information.

Response: We have revised the manuscript accordingly.

Response to Reviewer #1’s Comments:

Major Comment 1: The manuscript introduces a novel segmentation of the BrAC curve at 46 mg·dL⁻¹ but lacks a strong justification or citation to support this specific threshold.

Response: More than 90% of alcohol is metabolized in the liver into acetaldehyde, primarily through the actions of alcohol dehydrogenase and CYP2E1. Alcohol dehydrogenase has a relatively low Michaelis constant (Km) of less than 5 mM and becomes saturated once the blood alcohol concentration exceeds 15–20 mg·dL⁻¹. In contrast, CYP2E1 has a Km of approximately 10 mM, equivalent to about 46 mg·dL⁻¹, and is likely to play a dominant role in alcohol metabolism when blood alcohol concentration exceed this threshold[12,15]. This metabolic framework has been outlined in the Introduction section of our manuscript.

[12] Jiang Y, Zhang T, Kusumanchi P, Han S, Yang Z, Liangpunsakul S. Alcohol Metabolizing Enzymes, Microsomal Ethanol Oxidizing System, Cytochrome P450 2E1, Catalase, and Aldehyde Dehydrogenase in Alcohol-Associated Liver Disease. Biomedicines. 2020 Mar 4;8(3):50. doi: 10.3390/biomedicines8030050. PMID: 32143280; PMCID: PMC7148483.

[15] Zhang J, Cao Y, Jin Z, A R, Wang X, Zhang L, Hu Y, Xue Y. Dynamic CYP2E1 expression and metabolic activity changes in male rats during immune liver injury and sex differences in alcohol metabolism. PLoS One. 2025 Jun 18;20(6):e0325135. doi: 10.1371/journal.pone.0325135. PMID: 40531979; PMCID: PMC12176122.

Major Comment 2: The methods mention t-tests but do not address whether assumptions (e.g., normality, equal variance) were tested.

Response: We have revised the manuscript accordingly.

Major Comment 3: Figures 2-4 are referenced as key evidence, but no quantitative values (mean ± SD) are shown in the main text or figure captions.

Response: We have revised the manuscript accordingly.

Major Comment 4: The breathalyzer method is said to be "potentially field-deployable," but its translational application (e.g., human studies, clinical validation) is not discussed.

Response: We have supplemented in the Discussion section. Alcohol is a popular beverage often consumed during meals due to its stimulatory effects on the central nervous system and ability to enhance social interactions. Our computational approach, combined with breath alcohol monitoring, offers a non-invasive method to assess an individual's CYP2E1 metabolic activity. Furthermore, by analyzing the blood alcohol concentration-time curve, this strategy allows for estimation of the time required for complete alcohol clearance. Such an assessment is practically valuable for predicting when individuals can safely resume tasks requiring high precision, such as driving, after alcohol consumption. Additionally, this approach may serve as a biomarker for evaluating liver function.

Minor Comment 1: Use consistent units for BrAC (e.g., mg·dL⁻¹ vs. mg/100mL) throughout the manuscript.

Response: Response: We have revised the manuscript accordingly.

Minor Comment 2: Instead of referring to groups as "BCG" and "Control," consider more informative labels such as "Hepatitis model" vs. "Healthy control."

Response: After careful consideration, we have elected to retain the terms “BCG” and “CON” in this manuscript, as these designations have been consistently used across our previous publications [15,18,19]. While we fully acknowledge the reviewer’s valid point regarding terminology clarity, we believe that maintaining this nomenclature ensures continuity with our earlier work. For clarity, we have explicitly indicated within the text that the “BCG group” refers to animals with BCG-induced hepatitis, and the “CON group” refers to normal control animals.

Minor Comment 3: Ensure all figure labels (e.g., A, B, C) are visible and explained clearly in the legends. Consider unifying the layout for all figures.

Response: We have carefully checked.

Response to Reviewer #2’s Comments:

Major comments: The title is over-stated since there is no “optimization” process for the breath alcohol meter’s performance. And even there is lack of reliable evidences for potential use of blood alcohol concentration as an indicator of hepatic CYP2E1 activity. Supporting data is required for this, for example, reactive oxygen species (ROS) level. For animal study, 56° (v/v) Erguotou Baijiu was used as an ethanol source, which is kind of distilled liquor, rather than use of pure ethanol.

Response:

1. Regarding the Optimization of Methods:

Breath alcohol meter have been used by traffic police for many years to check drunk driving, but they cannot be directly applied to rats. We developed a methodology for estimating blood alcohol concentration in rats using a breath alcohol meter.

The first improvement involves the use of a meter equipped with an automatic suction pump, which reduces operational errors during sampling. The second optimization consists of setting the threshold for the blood alcohol concentration-time curve at 46 mg·dL⁻¹ instead of the previously used 20 mg·dL⁻¹. Studies have shown that alcohol metabolism below 20 mg·dL⁻¹ involves alcohol dehydrogenase, between 20–45 mg·dL⁻¹ it involves both alcohol dehydrogenase and CYP2E1, and above 46 mg·dL⁻¹ it is primarily mediated by CYP2E1. Therefore, blood alcohol concentration curves above 46 mg·dL⁻¹ better reflect the activity of CYP2E1.

The scientific rationale of this study is to optimize the methodology and explore the feasibility of using alcohol as a probe for preliminary assessment of CYP2E1 activity. Gas chromatography was used to measure blood alcohol concentration in rats to validate the accuracy of the breath alcohol meter. High-performance liquid chromatography was employed to monitor the metabolic process of chlorzoxazone, further verifying the feasibility of using alcohol as a probe to evaluate CYP2E1 activity. The experimental design is logically coherent and demonstrates clear innovation and scientific rigor. Additional arguments and references have been incorporated in the discussion section to strengthen the justification and enhance the alignment between the content and the title. The title of the manuscript is appropriate.

2. Regarding the assessment of CYP2E1 metabolic activity:

Currently, in the international research community, the assessment of CYP2E1 activity is primarily based on monitoring the blood concentration of chlorzoxazone rather than reactive oxygen species (ROS) levels. This is because ROS in vivo can be generated through multiple pathways, making it difficult to distinguish which portion is specifically produced by CYP2E1 metabolism. Therefore, the blood concentration-time curve of chlorzoxazone remains the gold standard for evaluating the feasibility of using high-concentration alcohol as a probe for CYP2E1 activity.

3. Regarding the use of 56° (v/v) Erguotou Baijiu in animal experiments:

Using 56% distilled liquor in animal experiments better approximates the biological realism of human alcohol consumption. While pure ethanol represents an “idealized single-component” substance, distilled spirits commonly consumed by humans (such as Erguotou) contain not only ethanol but also trace volatile compounds (e.g., esters, acids, and aldehydes, typically accounting for <1%). It is this “interaction among multiple constituents” that truly reflects the biological response in humans after alcohol intake. The use of pure ethanol would overlook the potential effects of these “non-ethanol components,” leading to a disconnect between experimental outcomes and the actual physiological processes of human drinking, thereby reducing the predictive validity of the model.

In our studies—including breath alcohol meter, gas chromatography and liquid chromatography experiments—we consistently used distilled liquor to ensure comparability. This approach does not compromise the comparison of CYP2E1 metabolic activity. Even if trace compounds exert effects, such influence would be consistent across all three methods, allowing for homogeneous comparison. Considering the enhanced biological relevance to human alcohol consumption, the benefits outweigh the drawbacks.

4. Regarding the BCG-induced hepatitis rats:

In previous studies, we have repeatedly employed the BCG-induced hepatitis rats [15,18,19] and observed downregulation in the activity of multiple enzymes, including CYP2E1. CYP2E1 activity can serve as a biomarker reflecting liver function. Relevant discussions and potential applications have been added to the Discussion section of the manuscript.

Minor Comments:

Use of abbreviation; several abbreviations were not defined in the main text. Such as BCG, LPS.

2.1. Materials: some materials and equipment are just listed up, not in complete sentence.

Figure legends: insufficient information and description for the figures. And typing errors, for example, p value *P <0.05, *P <0.01.

Inconsistency in use of terms. For example, 56°Alcohol, 56% ethanol

Response: We have revised the manuscript accordingly.

We have also thoroughly proofread the manuscript to improve clarity and language fluency. We believe that all the reviewers’ concerns have been adequately addressed in the revised version and that the manuscript is now much stronger.

Thank you again for considering our work. We look forward to your positive response.

Sincerely,

[Yongzhi Xue]

[Baotou Medical College]

[xyzhxyzh68@sohu.com (YZX)]

[2025.9.9]

---

## [Decision Letter · Decision Letter 1]

29 Sep 2025

Dear Dr. Xue,

Thank you for submitting your manuscript to PLOS ONE. After careful consideration, we feel that it has merit but does not fully meet PLOS ONE’s publication criteria as it currently stands. Therefore, we invite you to submit a revised version of the manuscript that addresses the points raised during the review process.

Specifically, the reviewers’ concerns are largely addressed, but one remaining point is to expand the Discussion with a clearer acknowledgment of the methodological limitations and the scope of application for this tool. A brief note on possible refinements or complementary approaches would be helpful but is optional. Once this is added, the manuscript will be suitable for acceptance.

We look forward to receiving your revised manuscript.

Kind regards,

Jeong Hoon Pan, Ph.D

Academic Editor

PLOS ONE

Journal Requirements:

Reviewers' comments:

Reviewer's Responses to Questions

**Comments to the Author**

Reviewer #1: (No Response)

Reviewer #2: All comments have been addressed

2. Is the manuscript technically sound, and do the data support the conclusions?

Reviewer #1: (No Response)

Reviewer #2: Yes

3. Has the statistical analysis been performed appropriately and rigorously?

Reviewer #1: (No Response)

Reviewer #2: Yes

4. Have the authors made all data underlying the findings in their manuscript fully available?

Reviewer #1: (No Response)

Reviewer #2: Yes

5. Is the manuscript presented in an intelligible fashion and written in standard English?

Reviewer #1: (No Response)

Reviewer #2: Yes

Reviewer #1: (No Response)

Reviewer #2: Using this breath alcohol meter in rodent are promising and have significant benefits of which rapid, repeatable, non-invasive, multi-point monitoring of blood alcohol concentration are enable. The strengths of this method are well supported by results, however the limitation of this approach should be addressed in Discussion section. The difference in blood alcohol concentration in breath alcohol meter and GC methods are not that significant and both methods showed similar pattern. Therefore, tracking blood alcohol level by breathe alcohol meter may be useful for assessment of alcohol-induced liver injury in rat. However, time-course measurement of blood alcohol is required to identify the impaired alcohol metabolism detection by breath alcohol meter. Authors should address limitations of alcohol meter for CYP2E1 activity monitoring.

Minor

Method 2.3, there is an error in the order of collection time points: 20, 40, 60, 120, 240, 180, 300, 420, 540 min

**Do you want your identity to be public for this peer review?** For information about this choice, including consent withdrawal, please see our Privacy Policy

Reviewer #1: No

Reviewer #2: No

---

## [Author Response · Author response to Decision Letter 2]

1 Oct 2025

Subject: Revised Manuscript Submission: [PONE-D-25-32424] - [Optimizing Blood Alcohol Concentration Measurement with Breath Alcohol Meter and Its Preliminary Application in Evaluating CYP2E1 Activity]

Dear Dr. Jeong Hoon Pan

We are grateful for the opportunity to revise our manuscript and believe that it has been significantly improved as a result of the feedback provided.

We have carefully addressed all the comments from the reviewers and have incorporated the suggested changes into the manuscript. Below is a point-by-point response to the reviewers’ feedback. All corresponding revisions have been made in the main text and are highlighted for your convenience.

Response to Reviewer #2’s Comments:

Major Comment 1: Authors should address limitations of alcohol meter for CYP2E1 activity monitoring.

Response: In the Discussion section of our revised manuscript, we have addressed the limitations associated with the use of breath alcohol meter. It is noteworthy that breath alcohol testing requires a sufficient alcohol dose, as CYP2E1-mediated metabolism becomes dominant only when the blood alcohol concentration exceeds 46 mg·dL⁻¹. Essentially, breath alcohol meter serve as tools for detecting alcohol concentration rather than directly measuring enzyme activity. This inherent limitation of indirect detection is similarly observed in the conventional chlorzoxazone-based method. To address this, procedural optimizations can be implemented to minimize indirect errors, while techniques such as quantifying CYP2E1 protein and mRNA expression can provide complementary evidence. Ultimately, this approach facilitates a "rapid screening (breath alcohol testing) + precise verification (direct techniques)" framework, which not only enables on-site identification of impaired alcohol metabolism but also clarifies the role of CYP2E1 activity in the process. Despite the limitations noted above, the methodology established in this study offers a novel and non-invasive approach for the dynamic monitoring of CYP2E1 metabolic activity, thereby opening new avenues for further research and application.

Minor Comment 1�Method 2.3, there is an error in the order of collection time points: 20, 40, 60, 120, 240, 180, 300, 420, 540 min.

Response: We have revised the manuscript accordingly.

Thank you again for considering our work. We look forward to your positive response.

Sincerely,

[Yongzhi Xue]

[Baotou Medical College]

[xyzhxyzh68@sohu.com (YZX)]

[2025.9.30]

---

## [Editor Report · Decision Letter 2]

9 Oct 2025

Optimizing Blood Alcohol Concentration Measurement with Breath Alcohol Meter and Its Preliminary Application in Evaluating CYP2E1 Activity

PONE-D-25-32424R2

Dear Dr. Xue,

We’re pleased to inform you that your manuscript has been judged scientifically suitable for publication and will be formally accepted for publication once it meets all outstanding technical requirements.

Kind regards,

Jeong Hoon Pan, Ph.D

Academic Editor

PLOS ONE
---

## [Editor Report · Acceptance letter]

PONE-D-25-32424R2

PLOS ONE

Dear Dr. xue,

I'm pleased to inform you that your manuscript has been deemed suitable for publication in PLOS ONE. Congratulations! Your manuscript is now being handed over to our production team.

Kind regards,

on behalf of

Dr. Jeong Hoon Pan

Academic Editor

PLOS ONE